# Effects of transboundary PM$_{2.5}$ transported from China on the regional PM$_{2.5}$ concentrations in South Korea: A spatial panel-data analysis

**Myung-Jin Jun** *, **Yu Gu**

Department of Urban Planning and Real Estate, Chung-Ang University, Seoul, Korea

* mjjun1@cau.ac.kr

## Abstract

This study identifies causal links between a high-PM$_{2.5}$ episode in Korea and air pollutants originating from China during a high-PM$_{2.5}$ episode that occurred in Korea between February 23 and March 12, 2019. Datasets on ground-based PM$_{2.5}$ levels in Korea and China, airflows from the back-trajectory models, and satellite images were investigated, and long-range transboundary transport (LRTT) effects were statistically analyzed using spatial panel-data models. The findings are: 1) visual presentations of the observed PM$_{2.5}$ concentration in China and Korea, back-trajectory air flows, and satellite images from the Moderate Resolution Imaging Spectroradiometer Aerosol Optical Depth and the Copernicus Atmosphere Monitoring Service clearly show that transboundary air pollutants from China affect PM$_{2.5}$ concentration in Korea; 2) the effect of LRTT from China is likely to intensify under certain meteorological conditions, such as westerly winds from China to Korea, the formation of high pressure in China and low pressure in Korea, relatively high temperature, and stagnant air flow in Korea; 3) the results from the spatial panel-data models provide statistical evidence of the positive effect of LRTT from China on increasing local PM$_{2.5}$ concentration in Korea. The nationwide average LRTT contributions to PM$_{2.5}$ concentration in Korea are 38.4%, while regional contributions are 41.3% for the Seoul Metropolitan Area, 38.6% for the northwest region, and 27.5% for the southeast regions in Korea, indicating the greatest impact on the Seoul Metropolitan Area.

## Introduction

Exposure to high concentrations of ambient particulate matter (PM), and specifically particles with a diameter of 2.5 μm or less (PM$_{2.5}$), can cause serious political and socio-economic effects globally, such as occupation and movement restrictions, productivity loss, and health issues, such as cardiovascular diseases, respiratory diseases, and asthma, all of which can lead to premature deaths [1–7]. The World Health Organization (WHO) announced that approximately 90% people breathe polluted air, contravening WHO air quality guidelines and

**Data Availability Statement:** Spatial Panel data is available from https://figshare.com/articles/dataset/Input_Data_Spatial_Panel/21699698.

**Funding:** The author(s) received no specific funding for this work.

**Competing interests:** The authors have declared that no competing interests exist.

resulting in 7 million premature deaths every year worldwide, 2.2 million of which occur in the Asia Pacific Region.

PM can either be directly released into the atmosphere or formed in the air as secondary particles from gases, such as sulfur dioxide ($SO_2$), nitrogen oxides ($NO_x$), and volatile organic compounds originating from anthropogenic sources, such as solid-fuel combustion for energy production, industrial activities, and vehicular exhaust or natural sources such as yellow dust [8]. As well as local emission sources and meteorological and chemical interactions, long-range transboundary transport (LRTT) is another cause of regional PM pollution [9, 10]. In Northeast Asia, LRTT of air pollution from China to South Korea (hereafter referred to as Korea) and Japan has been a serious issue as persistent westerly winds transport substantial amounts of air pollutants between these countries, particularly during winter [11–14]. The extent of LRTT of PM from China has worsened in recent decades as China has experienced rapid industrial development and a substantial growth in energy production from coal and oil combustion, contributing to increased emissions of particulate pollutants into the atmosphere [15–17].

Multiple researchers have investigated the effects of transboundary sources on regional PM concentrations among northeast Asian countries, and a wide range of LRTT impacts have been reported [18–23]. The North-East Asian Subregional Programme for Environmental Cooperation [18] reports the results of the source–receptor relationship (SRR) model, which estimates transboundary air pollution among eight source and receptor subregions: Northeast China, Northwest China, North China, Southern Southwest China, East China, North Korea, South Korea, and Japan. The SRR results show rates of self-contribution to $PM_{2.5}$ in South Korea and Japan of 47.4% and 39.4%, respectively, while LRTT contributions to $PM_{2.5}$ were 52.6% and 60.6%, respectively, mainly originating from Northeast China, North China, and East China. Kim et al. [19] noted that the $PM_{10}$ concentration in Seoul was majorly derived from China (39.8%–53.2%), domestic emissions in South Korea (15.4%–37.1%), and emissions from North Korea (9.0%–18.1%).

Kim et al. [20] reported similar results, arguing that the contribution of transboundary emissions from China was approximately 60% of the PM concentration in the Seoul Metropolitan Area in 2014, varying seasonally from 45% in September to 70% in March. Furthermore, Jeong et al. [21] used the potential source contribution function (PSCF) model to estimate the contribution of LRTT aerosols in East Asia to PM concentrations in Seoul, Korea from April 2007 to March 2008. Consequently, East and Northeast China, including Harbin and Chang-chun, the Pearl River Delta, Yangtze River Delta, and Beijing–Tianjin regions, were identified as the major source regions responsible for high $PM_{2.5}$ concentrations in Seoul. Lin et al. [23] used the Eulerian-type community multiscale air quality (CMAQ) model along with pollutant emissions and meteorological data from 2001 to measure the LRTT of acidifying substances in East Asia. They concluded that LRTT from East China contributes substantially (more than 20%) to anthropogenic reactive nitrogen and sulfur deposition in East Asia and demonstrates strong seasonal variation, generally peaking during the dry seasons. Kim [25] econometrically estimated the average effect of dust and industrial pollutant emissions from China on ambient pollution concentrations in South Korea using wind direction and speed from January 1, 2006, to December 31, 2014. It was found that southwest winds had the greatest effect throughout the year, accounting for 12.62% of the total annual mean $PM_{10}$ concentrations in South Korea.

Finally, the Korea National Institute of Environmental Research [22] collaborated with the United States National Aeronautics and Space Administration (NASA) to measure regional and hemispheric LRTT impacts on the Seoul Metropolitan Area using data collected from air-craft, ground stations, and ships between May and early June of 2016. They concluded that the $PM_{2.5}$ concentration in Seoul was highest during the short duration of direct transport of

$PM_{2.5}$ from China, with 48% of the $PM_{2.5}$ measured at Seoul Olympic Park originating from China and North Korea.

To evaluate the impact of LRTT, many researchers have employed the chemistry-transport model, community multi-scale air quality model [14, 24, 25], potential source contribution function method, concentration weighted trajectory model [21, 26, 27], aerosol optical depth (AOD) measurements, and back-trajectory models [14, 28, 29]. However, because the PM data used in these studies were derived from satellite AOD measurements or chemistry models, which are less precise than ground observations, the results were unreliable. Although previous studies have provided empirical evidence regarding the impact of LRTT from China on the PM concentrations in Korea and Japan, their findings have been debated due to the complex atmospheric chemistry, difficulty in identifying the causes of the $PM_{2.5}$ concentrations in receptor countries, and absence of an appropriate observation system and data with high spatial resolution [30–32].

The aim of this study is to assess the LRTT effects of $PM_{2.5}$ from China on the domestic $PM_{2.5}$ concentrations in South Korea during a high-$PM_{2.5}$ episode in Korea between February 23 and March 12, 2019, in which the concentration of $PM_{2.5}$ soared to 237 $\mu g/m^3$ and stayed at a national average of 55.7 $\mu g/m^3$, which is considered detrimental to health according to Korean air quality standards. In doing so, this study first attempts to identify causal links between a high-$PM_{2.5}$ episode in Korea and air pollutants originating from China by combining: 1) a hybrid single-particle Lagrangian integrated trajectory (HYSPLIT) backward model, 2) the $PM_{2.5}$ concentration observed in the monitoring stations of China and Korea, and 3) satellite images retrieved from the Moderate Resolution Imaging Spectroradiometer (MODIS) Aerosol Optical Depth (AOD) and the European Copernicus Atmosphere Monitoring Service (CAMS). In addition, the meteorological conditions favorable for maintaining the high-$PM_{2.5}$ episode in Korea are analyzed; for example, air pressure, wind direction and speed, and ambient temperature in both China and Korea. Finally, unlike previous studies, this study builds spatial panel-data models to statistically analyze the effects of the LRTT from China on regional $PM_{2.5}$ concentrations in Korea.

## Materials and methods

We collected ground-based $PM_{2.5}$ concentrations and meteorological datasets in Korea and China, together with remote sensing datasets and air flows from back-trajectory models to identify a causal association between a high-$PM_{2.5}$ episode in Korea and LRTT from China. The Chinese $PM_{2.5}$ data were obtained from the National Urban Air Quality Real-time Release Platform and the China National Environmental Monitoring Center, while the $PM_{2.5}$ data of Korea was sourced from Air Korea, Korea Environment Corporation. In addition, hourly local meteorological data, such as temperature, air pressure, and wind speed and direction, were collected from 95 monitoring stations in Korea, as the ambient $PM_{2.5}$ concentration is affected by meteorological factors. Fig 1 shows the locations of the air pollution ground monitoring stations in Korea (396 stations) and China.

To identify the spatiotemporal patterns of $PM_{2.5}$ transport from China to Korea, backward trajectories (calculated using the HYSPLIT model) were employed. As shown in Fig 1, nine major cities in Korea—Seoul, Busan, Daegu, Daejeon, Gwangju, Wonju, Chungju, Gangneung, and Jinju—were selected as the destination of backward trajectories.

Backward trajectories for 5 d and $PM_{2.5}$ concentrations observed in the source regions were calculated every 6 h from February 18 to March 12, 2019, as winds usually reach the Korean peninsula within 5 days (120 h of the total run time). A total of 92 GIS maps were generated from the HYSPLIT model for each location (23 days × 4 times per day). The NCEP and

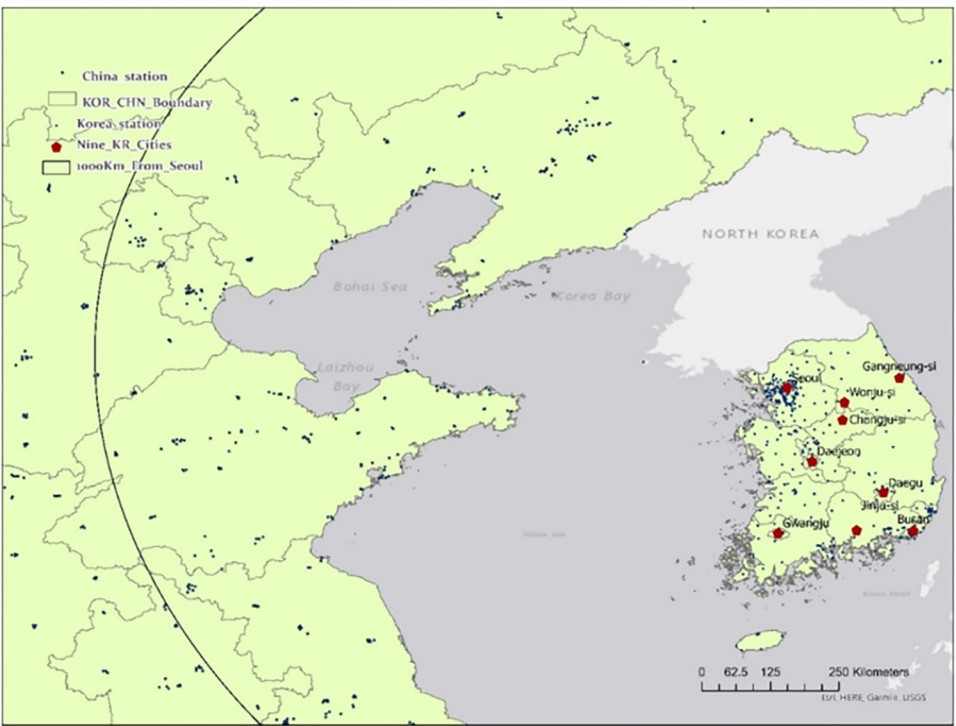

**Fig 1. Air pollution ground monitoring stations in Korea and China together with the nine Korean cities considered for back-trajectory models.**

National Centre for Atmospheric Research reanalysis dataset was used to assess the meteorological conditions in the backward trajectory calculations. Because the HYSPLIT model results provided 5 d spatiotemporal information (hourly x–y coordinates) about air transport pathways to each receptor city, identifying the source regions and departure time of air mass to the nine receptor cities in Korea was possible. Thus, the observed $PM_{2.5}$ concentrations at the departure time from the source region were used as the level of the LRTT-related $PM_{2.5}$ in this study.

Because remote sensing using satellite is known as a useful tool for tracking the LRTT of air pollutants [33–35], we also retrieved satellite images produced daily at 1 km pixel resolution from a MODIS Terra and Aqua combined Multi-angle Implementation of Atmospheric Correction (MAIAC) Land AOD gridded Level 2 product (https://lpdaac.usgs.gov/products/mcd19a2v006/). MODIS Terra and Aqua products were obtained from the Google Earth Engine (https://earthengine.google.com/). Data was also collected on the $PM_{2.5}$ concentration and wind direction obtained from the European CAMS (https://earth.nullschool.net/).

### Analysis of a causal link between high-$PM_{2.5}$ concentrations in Korea and long-range transboundary transport-related $PM_{2.5}$ from China

First, to estimate LRTT-related $PM_{2.5}$ values, we analyzed how the levels of $PM_{2.5}$ observed in Chinese cities on the back-trajectory pathways changed in the approach to the nine receptor Korean cities. Since we have information on the x-y coordinates of the Chinese cities in the pathways and the receptor cities, and on the $PM_{2.5}$ concentrations observed in Chinese cities at the passage time, it is possible to calculate the levels of $PM_{2.5}$ in Chinese cities by distance to the receptor cities in Korea during the study period. Fig 2 presents weekly average $PM_{2.5}$

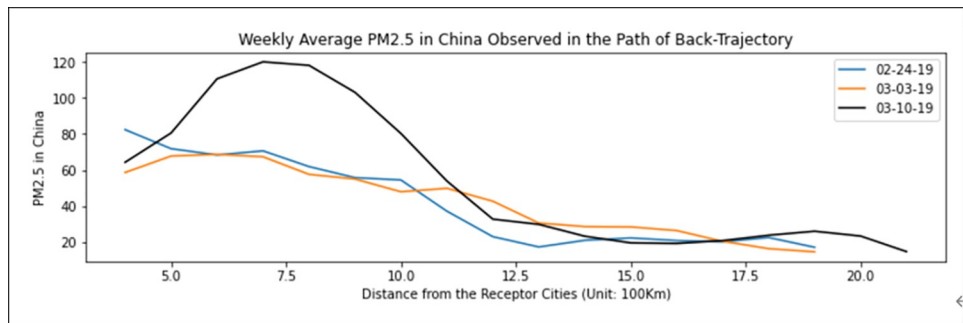

**Fig 2. Weekly average PM$_{2.5}$ concentrations observed in Chinese cities in the pathways of backward trajectories by the distance to the receptor cities in Korea.**

concentrations observed in Chinese cities in the pathways by distance to the receptor cities. This clearly shows that the PM$_{2.5}$ concentration starts to increase from around 1,200 km, worsening the closer it gets to the receptor cities from a distance of 1,000 km. This is because large cities with a high PM$_{2.5}$ concentration (such as Beijing, Tianjin, Shenyang, Dalian, and Qingdao, located in the northeastern region of China) that accommodate air pollution sources such as industrial factories, power plants, and heavy traffic volume are within a range of 1,000 km from Korea, as shown in Fig 1. Fig 2 indicates that as the air trajectories pass through these large cities, the LRTT PM$_{2.5}$ concentrations have increased. Therefore, this study uses the average PM$_{2.5}$ concentration observed at the passage time in Chinese cities in the pathways identified by the HYSPLIT backward trajectories within 1,000 km from the receptor cities as the LRTT-related variable in the following analysis, which is obtained every 6 h according to the arrival time at the receptor cities.

Fig 3 presents comparisons between the 12 *h* moving average PM$_{2.5}$ concentrations in Korea and the PM$_{2.5}$ levels observed in Chinese cities on the back-trajectory pathway during the study period and 1 week before and 1 week after the study period. Fig 3(A) indicates that Korea's PM$_{2.5}$ level was poor (36–75 μg/m$^3$) during the period 21–23 February and then deteriorated to a very bad level (76 μg/m$^3$ or above) from February 28, reaching the first peak on March 1. Following this, the second peak continued from March 4 to 7. The LRTT level from China also shows a similar pattern, with a change in the PM$_{2.5}$ level in Korea. The first and second peaks appear during the periods 21–23 and 25–26 February, and the third peak occurred during March 4–7. This suggests that the level of PM$_{2.5}$ concentration in Korea is highly associated with LRTT from China.

To analyze whether there are regional variations in Korea in terms of the association between PM$_{2.5}$ levels in Korean cities and LRTT PM$_{2.5}$ from China, nine receptor cities in Korea were grouped into three regions: 1) the Seoul Metropolitan Area (Seoul), 2) the northwest region (Wonju, Chungju, and Daejeon), and 3) the southeast region (Busan, Daegu, Gwangju, Gangneung, and Jinju). The remaining graphs show the level of PM$_{2.5}$ concentrations in the three regions in Korea and the level of LRTT PM$_{2.5}$ coming from China. Fig 3(B) and 3(C) show the levels of PM$_{2.5}$ concentration in the Seoul Metropolitan Area and the northwest region, averaged over 152 and 87 air pollution monitoring stations, respectively, and the level of LRTT from China. These graphs also show a similar pattern between the regional PM$_{2.5}$ level and LRTT from China during the March 2–8 period, indicating that the regional PM$_{2.5}$ level during this period was greatly affected by LRTT from China. Fig 3(D) shows that the PM$_{2.5}$ level in the southeastern region was relatively low during the March 2–8 period, possibly due to being a greater distance from China.

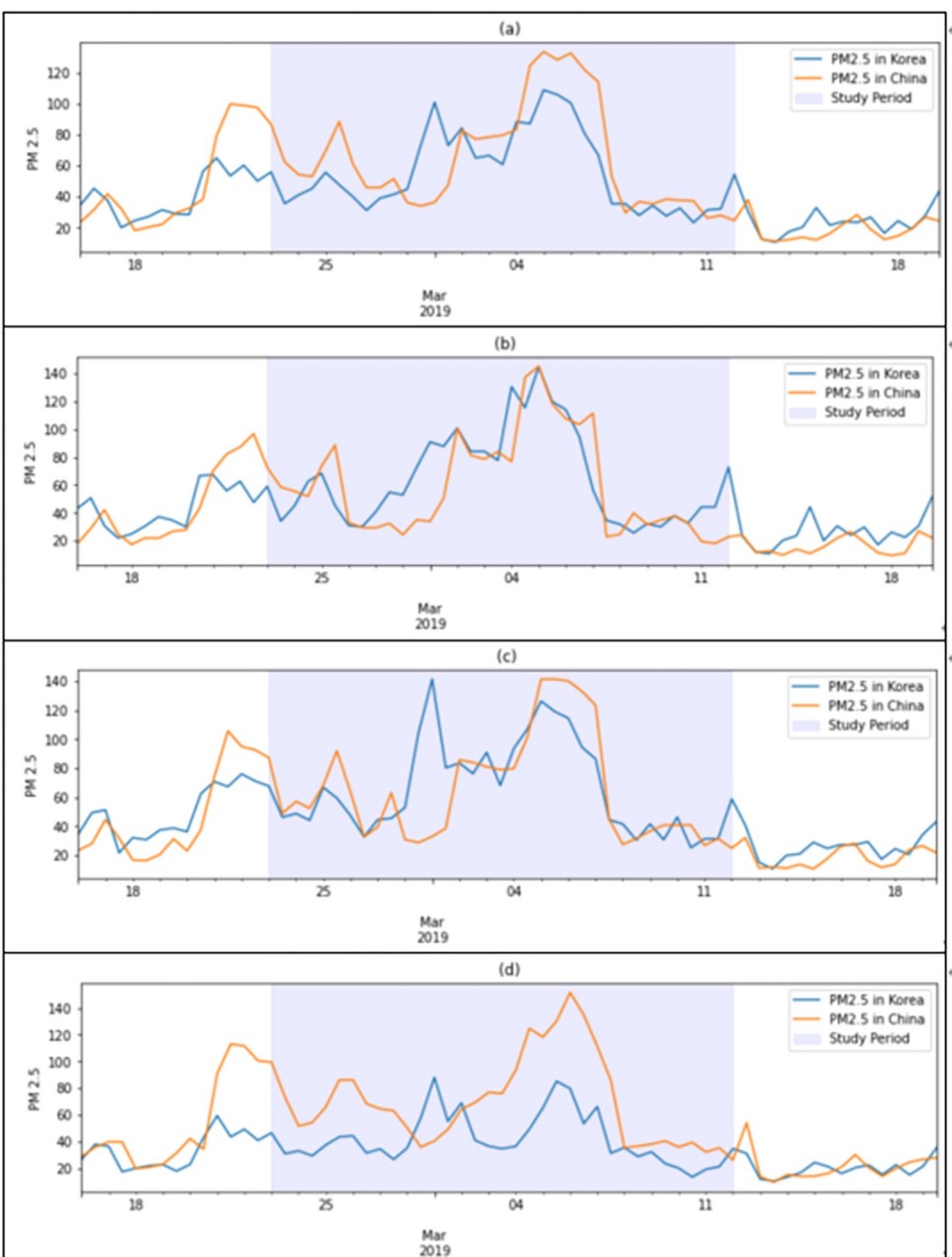

**Fig 3. PM$_{2.5}$ concentrations (μg/m$^3$) observed in Korean and Chinese cities on the back-trajectory pathway.**

## Synoptic meteorological conditions during the high-PM$_{2.5}$ episode

Because weather conditions also have a significant influence on the local PM$_{2.5}$ level, we investigated atmospheric pressure and wind direction and speed during high PM$_{2.5}$ periods. Fig 4 (A) shows the average wind direction obtained from the backward trajectory. This indicates that southwest and westerly winds prevail between 220–280˚ during the study period. Large cities such as Beijing, Tianjin, and Qingdao in the northeast of China are located in this direction from Korea. Fig 4(B) shows atmospheric pressure observed in Korea and China. Air pressure in China was observed in Chinese cities on the back-trajectory pathway. Relatively high

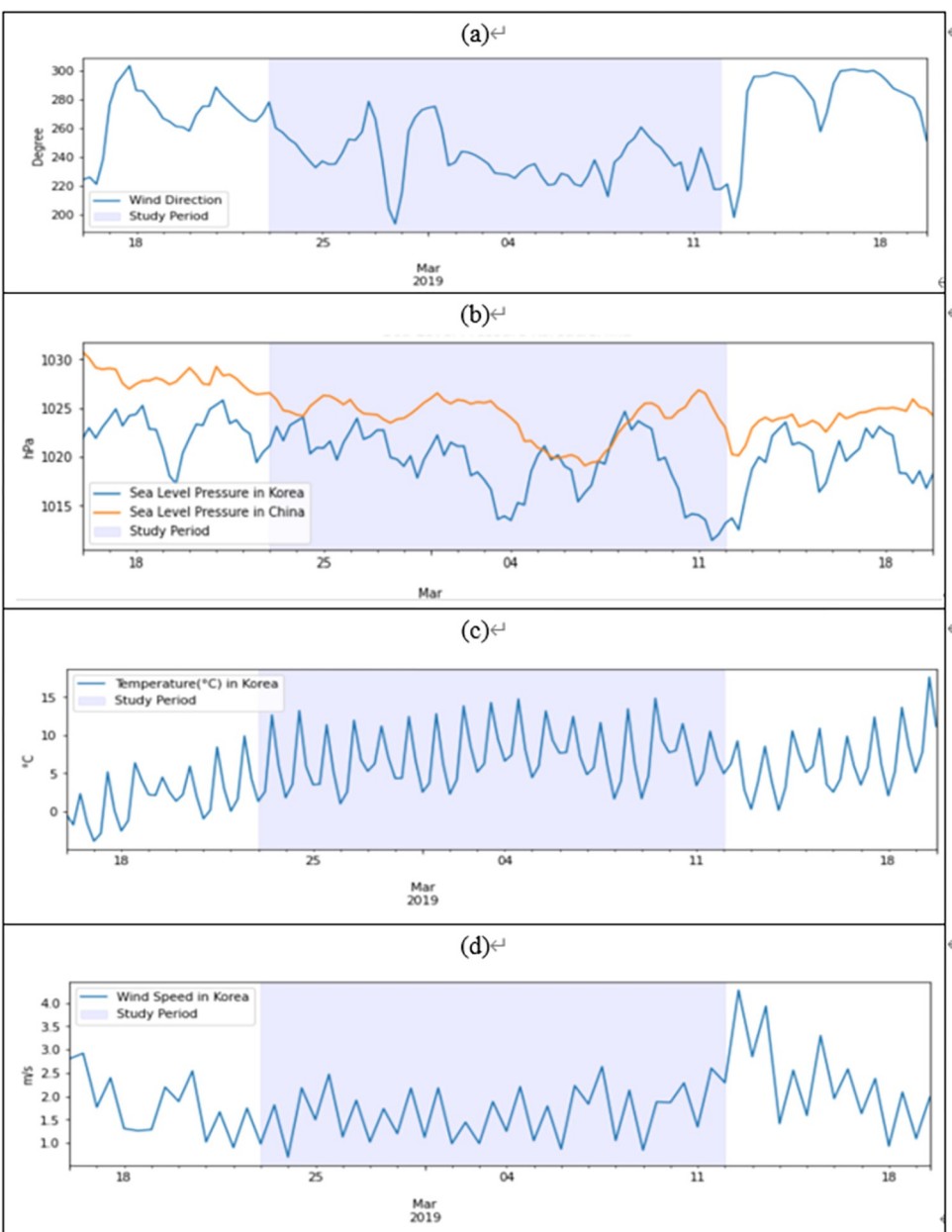

**Fig 4. Synoptic meteorological conditions during the high-PM$_{2.5}$ episode.**

air pressure was observed in China between 1020–1025 hPa, while air pressure in Korea was significantly lower, implying that winds are moving from areas of high pressure to areas of low pressure.

Fig 4(C) and 4(D) display temperature and wind speed observed in Korea. The ambient temperature remained relatively high (3–13°C) in spite of the winter season, while a stagnant and slow air flow prevailed at a speed of approximately 1.5 m/s, compared to that before and after the study period. This meteorological data indicates that westerly winds from China to Korea, the formation of high air pressure in China and low pressure in Korea, and relatively high temperature and stagnant air flow in Korea are considered to be the cause of high PM$_{2.5}$

LRTT from China and high PM$_{2.5}$ concentrations in Korea. Through analysis of synoptic meteorological conditions, we found that LRTT PM$_{2.5}$ from northeastern China contributed to the high-PM$_{2.5}$ episode in Korea when certain weather conditions were met. To reinforce this argument, the movement of air pollutants from China to Korea was examined using visual presentation of the spatial distribution of observed PM$_{2.5}$ concentrations in Korea and China, air flows of back-trajectories, and satellite image observations.

## Observed PM$_{2.5}$ concentrations in China and Korea and backward trajectory pathways

To examine the level of PM$_{2.5}$ concentrations in the Chinese regions on back-trajectory pathways, as shown in Fig 5, we mapped the observed PM$_{2.5}$ levels in China and Korea, and the back-trajectory pathways in terms of arrival time in the Korean peninsula obtained from the HYSPLIT model. The sample maps of the observed PM$_{2.5}$ concentrations in China show a seriously high PM$_{2.5}$ concentration exceeding 150 μg/m$^3$ in areas of northeastern China (such as Beijing, Tianjin, Dalian, and Qingdao). Conversely, the observed PM$_{2.5}$ concentrations in Korea were found to vary by region and time. For example, on February 23, the concentration of PM$_{2.5}$ was high in the southwestern part of Korea, while on March 1, a high PM$_{2.5}$ concentration prevailed throughout the Korean Peninsula. On March 4, a high PM$_{2.5}$ concentration prevailed in the western part of Korea, but on March 8, the concentration of PM$_{2.5}$ in Korea reduced to a low level.

The second column in Fig 5 depicts the sample maps of the 5 d backward trajectories to the nine receptor cities in terms of arrival time. These maps indicate the dynamics of air trajectories, showing variations in the source regions in China. Specifically, the back-trajectory path on February 23 and March 8 passed through cities located in northern China, such as Shenyang, Changchun, and Dalian, suggesting that Korean PM$_{2.5}$ levels were affected by LRTT from these regions. By contrast, looking at the maps depicting March 1 and 4, it was found that the back-trajectory path passed through areas with very high PM$_{2.5}$ concentrations, such as Beijing, Tianjin, Dalian, and Qingdao in China. These findings imply that PM$_{2.5}$ is transported from source regions over northeastern China, where many industrial factories and power plants are concentrated, to the nine receptor cities throughout the study period.

## Aerosol optical depth and copernicus atmosphere monitoring service PM$_{2.5}$ concentration with wind direction

This study visually inspected the satellite images retrieved from the MODIS-AOD and CAMS to reinforce the prior findings on the causal relationship between LRTT from China and a high-PM$_{2.5}$ episode in Korea. The first column in Fig 6 depicts sample maps on the daily median MODIS-AOD map, which presents the spatial distribution of AOD depth at 0.47 μ (AOD 470 nm), while the second column illustrates the CAMS PM$_{2.5}$ concentration with wind direction at 19:00 on the same day. The AOD maps show a high concentration of aerosols over the northeast and eastern regions of China during the sample period, similar to those observed PM$_{2.5}$ concentrations as shown in Fig 5. In addition, maps showing CAMS PM$_{2.5}$ concentrations with wind direction also indicate the movement path of PM$_{2.5}$. On February 23, LRTT PM$_{2.5}$ moved from the north to the south of the Korean Peninsula, while on March 1 and 4, PM$_{2.5}$ moved from the northeastern part of China to Korea following the westerly wind, which is consistent with the back-trajectory path in Fig 5. These satellite images show that westerly winds in and around the high AOD 470 regions play an important role in the formation of LRTT of PM$_{2.5}$ toward Korea.

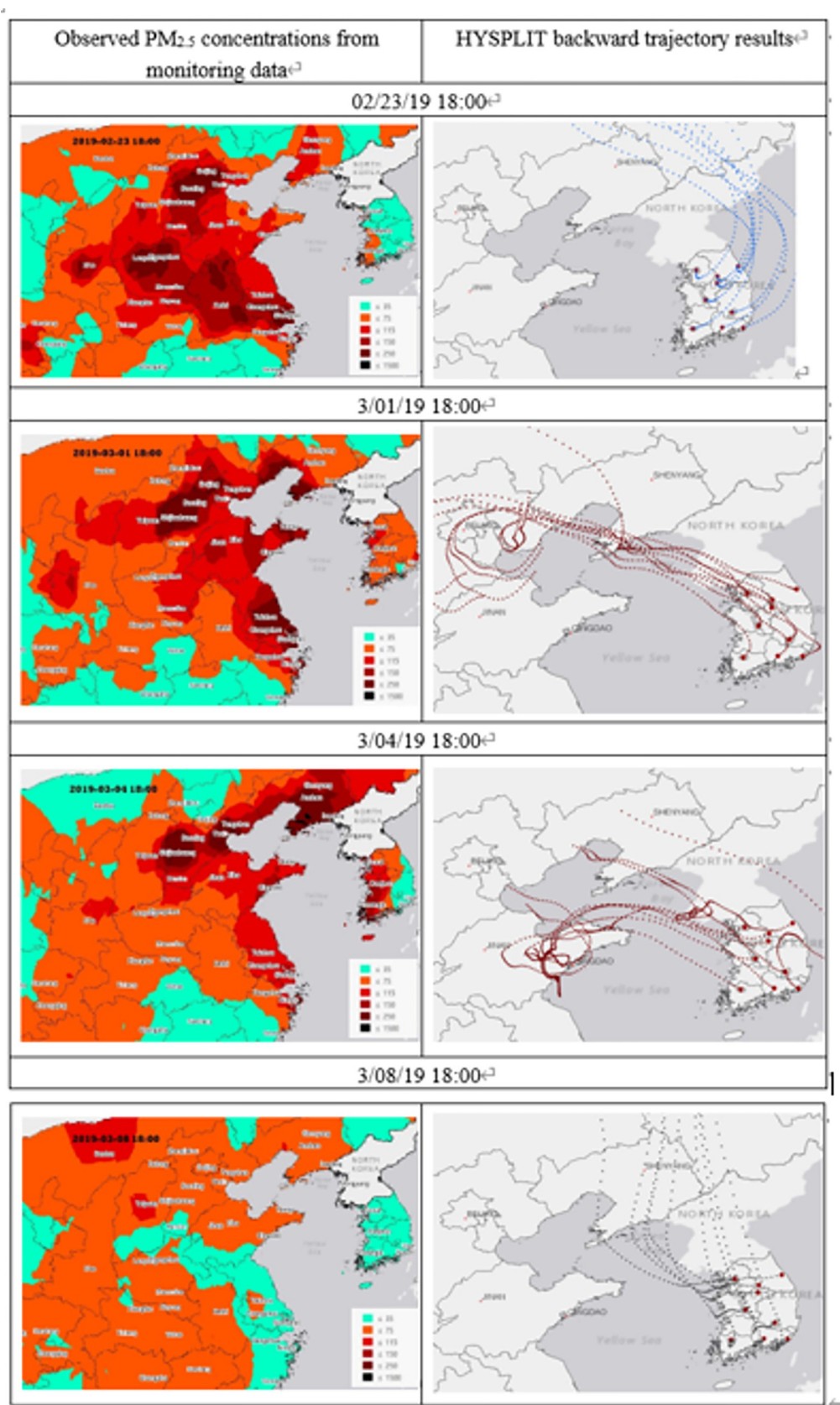

**Fig 5. Observed PM$_{2.5}$ concentrations from monitoring data and backward trajectory pathways.**

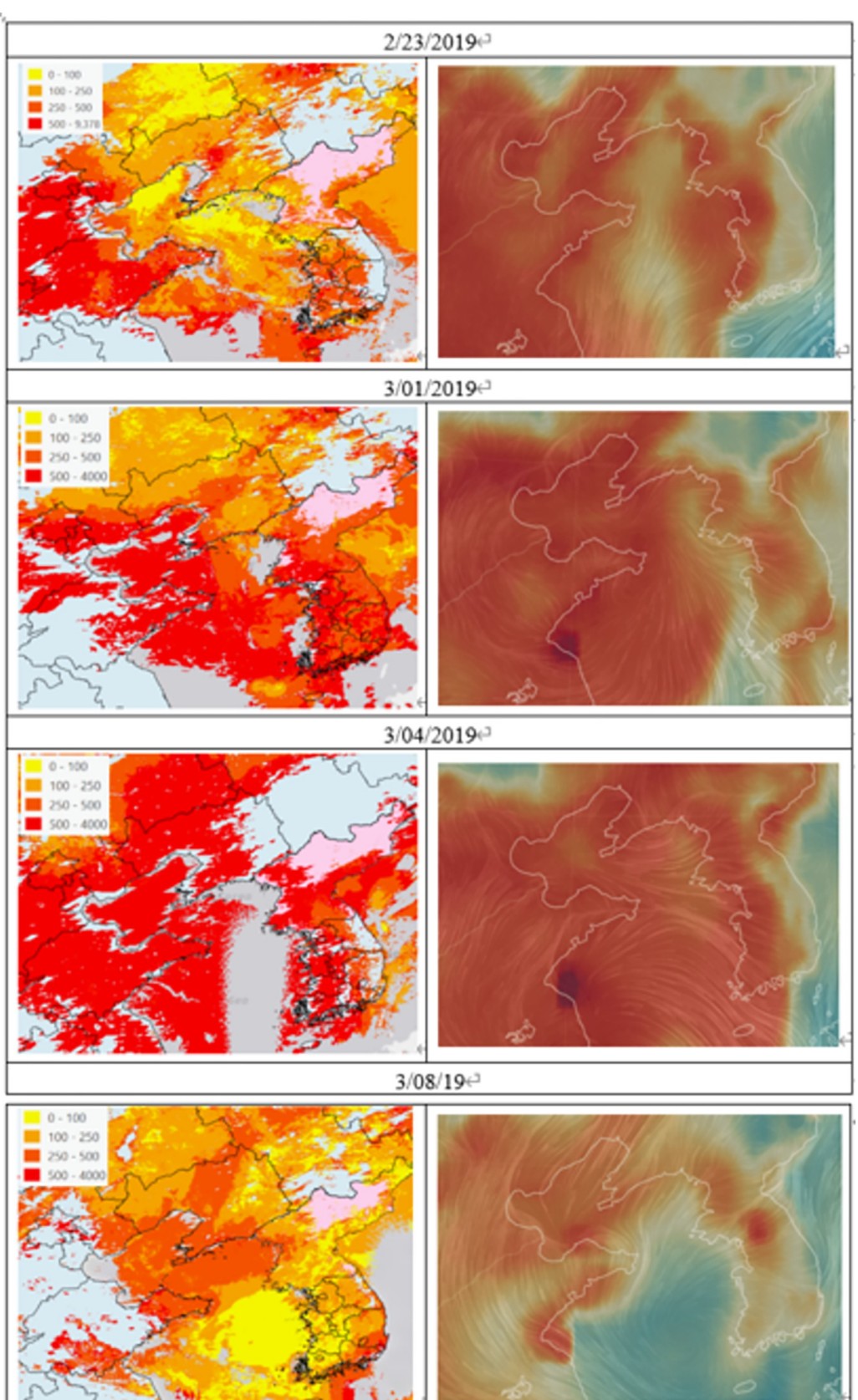

**Fig 6. Sample maps for aerosol optical depth and copernicus atmosphere monitoring service PM$_{2.5}$ concentration with wind direction.**

## Spatial panel-data models

From the data visualization on airflows through HYSPLT back-trajectory analysis, spatial distribution of PM$_{2.5}$ concentrations observed at monitoring stations in China and Korea, analysis on synoptic meteorological conditions, and satellite images retrieved from MODIS AOD and CAMS, it appears that LRTT of PM$_{2.5}$ from China contributed to the high-PM$_{2.5}$ episode in Korea. However, this analysis is not sufficient to statistically measure the extent to which LRTT from China has affected PM$_{2.5}$ levels in Korea.

Panel-data models were used to statistically estimate LRTT effects on PM$_{2.5}$ concentrations in Korea because the data consisted of a group of cross-sectional units (396 monitoring stations in Korea) observed during the study period. Panel data is more informative, shows high variability, low collinearity among dependent variables, high degrees of freedom, and high efficiency in estimation, while controlling for individual heterogeneity [36]. Multiple methods exist to fit the panel-data model depending on the assumptions of the explanatory variables, individual effects, errors in observed concentrations, and their relationships [37]. Two individual-specific effect models—the fixed effects (FE) model and the random effects (RE) model—assume that unobserved heterogeneity exists across individuals. Based on whether the individual-specific effects ($v_i$) are correlated with one or more of the explanatory variables, either an FE or RE model can be selected. If a correlation is observed, the FE model is used as it determines fixed individual effects over time. Conversely, if a correlation is not observed, the RE model is selected by considering the individual effects as random variables over time.

The Hausman test was conducted to select the appropriate model using *Stata* [38], even though this is only usually used for static models. The Hausman test is an endogeneity test that assumes a null hypothesis of zero covariance between the individual effects and independent variables. In this study, the null hypothesis was rejected at the 1% significance level for both models, suggesting the suitability of the FE model. We built four spatial panel FE models in this study to analyze nationwide and three local (Seoul Metropolitan Area, northwest region, and southeast region) effects of LRTT. In the spatial panel-data model, the dependent variables were the natural logarithm of observed PM$_{2.5}$ concentrations taken at four times each day at 396 monitoring stations in Korea, while the predictor variables were LRTT of PM$_{2.5}$ transported from the China, secondary precursors such as SO$_2$ and nitrous dioxides (NO$_2$) observed at Korean monitoring stations, and meteorological variables, including ambient temperature, rainfall, pressure, humidity, and wind speed and direction, observed at the Korean weather monitoring stations. For the wind direction, three westerly wind directional dummy variables are included: west-southwest (WSW: 1 if the azimuth wind direction is between 213 and 281 and 0 otherwise), 2) west-northwest (WNW: 282 and 326), and 3) north-northwest (NNW: 327 and 360). In addition, we added two lag variables of NO$_2$ and SO$_2$, which are key determinants of the regional PM2.5 concentration as the secondary precursor to consider serial dependencies.

Table 1 presents descriptive statistics for the variables used in the four spatial panel models. Average ambient temperature, humidity, and pressure in Korea were 7.1˚C, 59.4%, and 1009.8 hPa, respectively. The local wind speed was 1.72 m/s. Because South Korea is in the westerly zone, three westerly wind dummy variables account for 67% of the total wind direction. Because the number of PM$_{2.5}$ observation stations was different in each region, each dataset had a different panel length, ranging from 87 (northwest) to 157 (southeast). The nationwide average hourly PM$_{2.5}$ concentration was 55.7 μg/m$^3$, while the average PM$_{2.5}$ concentration of

**Table 1. Descriptive statistics for the variables used in the spatial panel-data model.**

| | | All Korea | | Seoul Metropolitan Region | | Northwest Region | | Southeast Region | |
|---|---|---|---|---|---|---|---|---|---|
| | | Mean | Std. | Mean | Std. | Mean | Std. | Mean | Std. |
| Hourly $PM_{2.5}$ in South Korea ($\mu g/m^3$) | | 55.70 | 36.41 | 64.74 | 38.09 | 65.31 | 38.32 | 41.62 | 28.23 |
| $SO_2$ (ppm) | | 4.E-03 | 3.E-03 | 0.01 | 3.E-03 | 4.E-03 | 2.E-03 | 4.E-03 | 3.E-03 |
| Lag of $SO_2$ (ppm) | | 4.E-03 | 3.E-03 | 0.01 | 3.E-03 | 4.E-03 | 2.E-03 | 4.E-03 | 3.E-03 |
| $NO_2$ (ppm) | | 0.03 | 0.02 | 0.04 | 0.02 | 0.03 | 0.01 | 0.02 | 0.01 |
| Lag of $NO_2$ (ppm) | | 0.03 | 0.02 | 0.04 | 0.02 | 0.03 | 0.01 | 0.02 | 0.01 |
| Ambient Temperature (℃) | | 7.21 | 4.37 | 6.29 | 4.28 | 6.09 | 4.66 | 8.73 | 3.83 |
| Rainfall (mm) | | 0.08 | 0.68 | 0.01 | 0.14 | 0.02 | 0.17 | 0.17 | 1.05 |
| Local Wind Speed (m/s) | | 1.73 | 1.38 | 1.74 | 1.19 | 1.23 | 1.06 | 2.01 | 1.60 |
| Wind Direction Dummy | WSW (213-281˚) | 0.22 | 0.42 | 0.26 | 0.44 | 0.21 | 0.40 | 0.19 | 0.39 |
| | WNW (282-326˚) | 0.18 | 0.38 | 0.21 | 0.41 | 0.15 | 0.36 | 0.15 | 0.36 |
| | NNW (327-360˚) | 0.27 | 0.45 | 0.21 | 0.41 | 0.37 | 0.48 | 0.27 | 0.45 |
| Humidity (%) | | 59.09 | 21.91 | 54.66 | 23.04 | 61.12 | 22.67 | 62.26 | 19.53 |
| Air Pressure (hPa) | | 1009.79 | 8.66 | 1010.63 | 6.05 | 1008.03 | 10.40 | 1009.95 | 9.57 |
| $PM_{2.5}$ from China ($\mu g/m^3$) | | 63.88 | 37.57 | 58.81 | 37.23 | 64.16 | 37.12 | 68.64 | 37.52 |
| Number of groups (N) | | 396 | | 152 | | 87 | | 157 | |
| Panel length (T) | | 71 | | 71 | | 71 | | 71 | |
| Total Observations | | 28,116 | | 10,792 | | 6,177 | | 11,147 | |

LRTT from China was 63.9 $\mu g/m^3$. By contrast, the average hourly $PM_{2.5}$ concentration by region was recorded as 64.7 $\mu g/m^3$ in the SMA, 65.3 $\mu g/m^3$ in the northwest region, and 41.6 $\mu g/m^3$ in the southeast region, indicating that the concentration of $PM_{2.5}$ in the southeastern region is relatively lower than in either the Seoul Metropolitan Area or the northwestern region. The average $PM_{2.5}$ concentrations of LRTT from China ranged from 58.8 $\mu g/m^3$ to 68.6 $\mu g/m^3$, depending on the receptor city in Korea.

To check the spatial autocorrelation of the four panel datasets, three diagnostic tests—Moran's I statistics for residuals, Lagrange multiplier (LM), and robust LM tests for lag and error dependence—were performed to determine the existence of spatial dependence in the panel-data models. Subsequently, as shown in Table 2, the results of analysis rejected the hypothesis that there were no spatially lagged dependent variables and no spatially autocorrelated error terms at the 1% significance level for all four panel datasets.

The selection of an appropriate spatial weight matrix is an important step for building spatial econometric models because it represents the spatial relationship between observations. Because there is no rule of thumb for choosing an appropriate weight matrix, it is necessary to check model robustness with different spatial weight matrices based on spatial patterns and

**Table 2. Moran's I statistics for residuals, LM, and robust LM test results.**

| Test Type | All Korea | Seoul | Northwest | Southeast |
|---|---|---|---|---|
| Moran's I (error) | 0.166*** | 0.18*** | 0.084*** | 0.07*** |
| Lagrange Multiplier (lag) | 2947.27*** | 43.71*** | 150.58*** | 241.41*** |
| Robust LM (lag) | 53.17*** | 93.91*** | 5.064** | 13.02*** |
| Lagrange Multiplier (error) | 5692.98*** | 2478.56*** | 316.31*** | 398.20*** |
| Robust LM (error) | 2798.89*** | 2528.76*** | 170.79*** | 169.81*** |

Note: *** $p < 0.01$, ** $p < 0.05$

data structures [39]. Accordingly, we compared the Akaike Information Criteria (AIC) for the spatial error model (SEM) and the spatial autoregressive model (SAR) using various spatial weighting matrices to find the best model. We specifically tested seven different spatial weight matrices for the SEM and SAR models: queen contiguity, three distance-based (2, 4, and 8 k-nearest-neighbors (KNN)) and three kernel functions (uniform, triangular, and gaussian). We found the AIC ranging from 554,943 for the 8-KNN to 558,068 for the triangular kernel function in the SAR and from 554,850 for the 8-KNN to 558,763 for the triangular kernel function in the SEM model. We finally selected the SEM models with the 8-KNN spatial weight matrix as the most appropriate for this study as they have the lowest AIC.

The SEM model takes the following mathematical forms [40]:

$$Y_{Nt} = \alpha l_{Nt} + X_{Nt}^k \beta^k + u \tag{1}$$

$$u = \lambda W u + \varepsilon$$

where $Y$ is an $N \times t$ vector of the dependent variable (N = 396 $PM_{2.5}$ observation stations for the national model, and t = 71 times (18 days $\times$ 4 h minus 1 due to the addition of the lag variables); $l_{Nt}$ is an $N \times t$ vector; $X$ denotes an $Nt \times k$ matrix of explanatory variables (k = 13 including two lag variables on $NO_2$ and $SO_2$); $\alpha$, $\beta$ are the parameters to be estimated; $\varepsilon$ and $u$ are vectors of error terms; $W$ is an $N \times N$ spatial weight matrix; and $\lambda$ denotes the autocorrelation coefficients.

## Spatial panel-data model results

As described in the previous section, four spatial panel-data FE models were developed based on LRTT receptor regions. Table 3 shows the spatial panel-data model results. The pseudo $R^2$ indicates that the FE estimators of the four models explained 28%–56%. The significant values of $\lambda$ reflect the existence of spatially correlated errors for all four models. Results show that high values of $NO_2$ and $SO_2$, and their lag variables were positively related with high values of local $PM_{2.5}$ concentrations for all estimators at the 1% significance level. In relation to meteorological variables, rainfall was negatively related with the level of local $PM_{2.5}$ concentration, while ambient temperature and humidity had a positive association (except for the southeast model). There was a negative correlation with wind speed, indicating that the higher the wind speed, the lower the local $PM_{2.5}$ concentration. The wind direction dummy variables have different effects on the local $PM_{2.5}$ concentration. The west-northwest (WNW) has a statistically significant positive effect for all four models, while the north-northwest (NNW) has statistically insignificant or negative coefficient values. This finding shows statistically that the local $PM_{2.5}$ concentration is affected by the $PM_{2.5}$ level in Chinese cities located in the northeastern region of China (such as Beijing, Tianjin, and Dalian).

The coefficient value for the association of air pressure with the $PM_{2.5}$ concentration in the Seoul model was significantly negative, whereas it was positively correlated to the local $PM_{2.5}$ concentration in the southeast model. Most importantly, the LRTT variables provided significant positive coefficient values for all four models. These findings provide statistical evidence that LRTT from China has significant effects on increasing local $PM_{2.5}$ concentrations in Korea. A unit increase in LRTT from China was associated with a 0.6%, 0.7%, 0.6%, and 0.4% increase in the regional $PM_{2.5}$ concentration for all of Korea, the Seoul Metropolitan Area, the northwest region, and the southeast region, respectively, during the study period. The results of this analysis indicate that the Seoul Metropolitan Area had the greatest LRTT impact from China. It also suggests that the northwest region had a higher LRTT effect than the southeast region, indicating a decreasing LRTT impact as the distance from China increases.

**Table 3. Spatial panel-data FE model results.**

| Variable | | All Korea | Seoul Metropolitan Region | Northwest Region | Southeast Region |
|---|---|---|---|---|---|
| $NO_2$ | | 9.801*** | 8.683*** | 7.272*** | 13.818*** |
| Lag of $NO_2$ (ppm) | | 4.435*** | 2.382*** | 4.998*** | 6.840*** |
| $SO_2$ | | 26.909*** | 26.421*** | 55.903*** | 18.315*** |
| Lag of $SO_2$ (ppm) | | 7.706*** | 4.541*** | 20.257*** | 9.023*** |
| Rainfall (mm) | | −0.113*** | −0.149*** | −0.188*** | −0.070*** |
| Ambient temperature (℃) | | 0.019*** | 0.053*** | 0.024*** | −0.034*** |
| Wind speed (m/s) | | −0.015*** | −0.009** | −0.011 | −0.007* |
| Local Wind Direction | WSW | 0.066*** | −0.028*** | 0.025 | 0.108*** |
| | WNW | 0.092*** | 0.049*** | 0.074*** | 0.095*** |
| | NNW | −0.007 | −0.031*** | 0.024 | −0.027** |
| Humidity (%) | | 0.008*** | 0.016*** | 0.010*** | −0.004*** |
| Air pressure (hPa) | | −0.001** | −0.016*** | 0.001 | 0.006*** |
| $PM_{2.5}$ concentration from China ($\mu g/m^3$) | | 0.006*** | 0.007*** | 0.006*** | 0.004*** |
| λ | | 0.625*** | 0.567*** | 0.521*** | 0.565*** |
| Pseudo $R^2$ | | 0.333 | 0.559 | 0.357 | 0.282 |
| AIC | | 328425.5 | 110666.6 | 62238.7 | 120723.5 |
| BIC | | 328532.6 | 110761.3 | 62326.1 | 120818.7 |
| Number of groups (N) | | 396 | 152 | 87 | 157 |
| Panel length (T) | | 71 | 71 | 71 | 71 |
| N | | 28,116 | 10,792 | 6,177 | 11,147 |

*** p < 0.01

** p < 0.05

* p < 0.1

Table 4 presents the average effects of LRTT from China on the nationwide and three local $PM_{2.5}$ concentrations. Calculating the average effects of LRTT from China on the local $PM_{2.5}$ concentrations requires several steps. First, since we took a natural log of local $PM_{2.5}$ concentration as dependent variable, the percent change in local $PM_{2.5}$ level for a unit increase in the LRTT can be obtained from the coefficient of the LRTT. Second, the marginal effect of a unit increase in LRTT on the local $PM_{2.5}$ level can be computed by multiplying the percentage change by the Korean average $PM_{2.5}$ level and dividing by 100. Third, multiplying this by average LRTT yields the average LRTT effect on local $PM_{2.5}$ level. The average effects of LRTT from China were 21.4 $\mu g/m^3$, 26.8 $\mu g/m^3$, 25.2 $\mu g/m^3$, and 11.5 $\mu g/m^3$ on the nationwide, the Seoul Metropolitan Area, the northwest, and the southeast regions, respectively, which account

**Table 4. Average effects of LRTT from China on local $PM_{2.5}$ concentration.**

| | Nationwide | Seoul Metropolitan Area | Northwest Region | Southeast Region |
|---|---|---|---|---|
| LRTT Coefficient from the Spatial Panel FE Model | 0.006*** | 0.007*** | 0.006*** | 0.004*** |
| LRTT Mean $PM_{2.5}$ ($\mu g/m^3$) | 63.88 | 58.81 | 64.16 | 68.64 |
| Marginal effect of a unit increase in LRTT on local $PM_{2.5}$ level | 0.34 | 0.45 | 0.39 | 0.17 |
| Average LRTT effect on local $PM_{2.5}$ level | 21.41 | 26.75 | 25.22 | 11.45 |
| Local Mean $PM_{2.5}$ in South Korea ($\mu g/m^3$) | 55.70 | 64.74 | 65.31 | 41.62 |
| Percentage of LRTT effect on local mean $PM_{2.5}$ | 38.4% | 41.3% | 38.6% | 27.5% |

*** p < 0.01

for 38.4%, 41.3%, 38.6%, and 27.5% of average nationwide and regional $PM_{2.5}$ concentrations, respectively. These findings indicate that the Seoul Metropolitan Area, which is close to the northeastern part of China and where 50.3% of the nation's population resides (26.0 million in 2022), is most affected by $PM_{2.5}$ LRTT from China during the study period.

These LRTT effects were close to the range of 40%–60% suggested by previous studies. The LRTT effect was 52.6% according to NEASPEAC[18], 39.8%–53.2% as reported by Kim et al. [19], approximately 60% according to Kim et al. [20], and 48% according to the Korea National Institute of Environmental Research [22]. This study verified the findings of the previous studies, which argued that westerly winds transport substantial amounts of air pollutants from China to Korea during the winter [11–14]. Furthermore, along with LRTT effects, secondary formation of $PM_{2.5}$ from precursor pollutants, such as $NO_2$ and $SO_2$, was a key determinant of the regional $PM_{2.5}$ concentration.

## Conclusions

Despite the recent decades' efforts to curb domestic air pollution, Korea is exposed to frequent and serious air pollution during the winter and spring seasons, threatening public health and socioeconomic activities. It is believed that air quality in Korea is affected not only by domestic stationary and mobile sources but also by the LRTT of air pollutants from external sources. Focusing on the high-$PM_{2.5}$ episode during the period February 23–March 12, 2019, this study identified air flows from the highly polluted regions in the northeast of China to Korea through visual exploratory analysis using HYSPLT back-trajectory, spatial distribution of $PM_{2.5}$ concentration observed at air pollution monitoring stations in China and Korea, and satellite images retrieved from MODIS AOD and CAMS. In addition, meteorological conditions related to a high-$PM_{2.5}$ episode in Korea were analyzed, including wind direction and speed, air pressure, and ambient temperature. Finally, spatial panel-data models were built to statistically measure the effect of LRTT from China on the $PM_{2.5}$ level in Korea.

The findings can be summarized as follows. First, visual presentations of the observed $PM_{2.5}$ concentration in China and Korea, back-trajectory air flows, and satellite images from the MODIS-AOD and CAMS, clearly show that transboundary air pollutants from China affect $PM_{2.5}$ concentration in Korea. Second, the effect of LRTT from China is likely to be intensified with certain meteorological conditions, such as westerly winds from China to Korea as well as the formation of high pressure in China and low pressure in Korea and relatively high temperature and stagnant air flow in Korea. Third, the results from the spatial panel-data models provide statistical evidence on the positive effect of LRTT from China on increasing local $PM_{2.5}$ concentrations in Korea. The LRTT effects vary by region in Korea: the greatest impact was found to be on the Seoul Metropolitan Area with a decreasing impact as the distance from China increased.

Important implications and insights can be drawn from the empirical findings of this study. First, the results indicate that since approximately 40% of the domestic $PM_{2.5}$ concentration, on average, is contributed by LRTT from Chinese cities during high-$PM_{2.5}$ episodes, controlling the source emission release in China is essential to mitigating air pollution in Korea. Second, the fact that the SMA, where more than half of the nation's population resides, receives the greatest LRTT effect indicates that citizens in the Seoul Metropolitan Area likely suffer the most severe health damage from LRTT exposure. Active diplomatic efforts need to be implemented for international collaboration between Korea and China, while corresponding emission reduction policies should be developed. Third, since secondary formation from precursor pollutants such as $NO_2$ and $SO_2$ have significant impacts on domestic $PM_{2.5}$ concentrations, the release of these secondary precursors should be controlled and minimized to prevent the

exacerbation of air pollution and to prevent $PM_{2.5}$ from reaching detrimental concentrations. Future studies can focus on generalizing the LRTT effect by extending the current spatial panel-data approach with a longer study period and analyzing the spatiotemporal trends and seasonal and meteorological variations of LRTT contributions to local $PM_{2.5}$ concentrations for various high-$PM_{2.5}$ episodes in Korea.

## Author Contributions

**Conceptualization:** Myung-Jin Jun.

**Data curation:** Myung-Jin Jun, Yu Gu.

**Formal analysis:** Yu Gu.

**Methodology:** Myung-Jin Jun.

**Visualization:** Yu Gu.

**Writing – original draft:** Myung-Jin Jun.

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
