## [Decision Letter · Decision Letter 0]

27 May 2022

PONE-D-21-36719Effects of transboundary PM2.5 transported from Chinese cities on the regional PM2.5 concentrations in South Korea: A s patial panel-data analysisPLOS ONE

Dear Dr. JUN,

Thank you for submitting your manuscript to PLOS ONE. After careful consideration, we feel that it has merit but does not fully meet PLOS ONE’s publication criteria as it currently stands. Therefore, we invite you to submit a revised version of the manuscript that addresses the points raised during the review process.

We look forward to receiving your revised manuscript.

Kind regards,

Yangyang Xu

Academic Editor

PLOS ONE

Journal Requirements:

2. Please remove your figures from within your manuscript file, leaving only the individual TIFF/EPS image files, uploaded separately.  These will be automatically included in the reviewers’ PDF

3. We note that Figures 1, 3 and 4 in your submission contain [map/satellite] images which may be copyrighted. All PLOS content is published under the Creative Commons Attribution License (CC BY 4.0), which means that the manuscript, images, and Supporting Information files will be freely available online, and any third party is permitted to access, download, copy, distribute, and use these materials in any way, even commercially, with proper attribution. For these reasons, we cannot publish previously copyrighted maps or satellite images created using proprietary data, such as Google software (Google Maps, Street View, and Earth). For more information, see our copyright guidelines: http://journals.plos.org/plosone/s/licenses-and-copyright.

a. You may seek permission from the original copyright holder of Figures 1, 3 and 4 to publish the content specifically under the CC BY 4.0 license. 

Reviewers' comments:

Reviewer's Responses to Questions

**Comments to the Author**

1. Is the manuscript technically sound, and do the data support the conclusions?

Reviewer #1: Yes

Reviewer #2: No

2. Has the statistical analysis been performed appropriately and rigorously? 

Reviewer #1: Yes

Reviewer #2: No

3. Have the authors made all data underlying the findings in their manuscript fully available?

Reviewer #1: No

Reviewer #2: Yes

4. Is the manuscript presented in an intelligible fashion and written in standard English?

Reviewer #1: Yes

Reviewer #2: Yes

5. Review Comments to the Author

Reviewer #1: The author has claimed that some restrictions apply to the data availability. The author needs to process data for public access. For additional comments on the statistical analysis, please see the report attached.

Reviewer #2: Please refer to my attached report.

6. PLOS authors have the option to publish the peer review history of their article (what does this mean?). If published, this will include your full peer review and any attached files.

Reviewer #1: No

Reviewer #2: No

---

## [Author Response · Author response to Decision Letter 0]

6 Sep 2022

Dear Reviewer,

Thank you for your insightful comments and feedback on the manuscript, which have enriched the manuscript and produced a better and more balanced account of the research. The following revisions were made to fulfill the reviewers’ requirements. I believe they have added to the value and quality of the manuscript.

1. The revised version uses a back-trajectory model that identifies air flows from China to nine receptor cities in Korea, while the original version of the manuscript used the forward-trajectory model starting from 12 source cities in China to analyze the effect of transboundary PM2.5 transported from these cities on the PM2.5 concentrations in South Korea. As we changed from forward-trajectory to back-trajectory, the focus of our analysis also changed from LRTT effects from source cities in China to effects on receptor cities in Korea.

2. Along with the back-trajectory model and the PM2.5 concentration observed in the monitoring stations in China and Korea, we collected satellite images retrieved from the Moderate Resolution Imaging Spectroradiometer (MODIS) Aerosol Optical Depth (AOD) and the European Copernicus Atmosphere Monitoring Service (CAMS) to identify causal links of a high-PM2.5 episode in Korea with air pollutants originated from China.

3. We analyzed the meteorological conditions favorable for maintenance of the high-PM2.5 episode in Korea, such as air pressure, wind direction and speed, and ambient temperature in both China and Korea.

4. We have provided a detailed explanation on determining PM2.5 levels originating from China. We found that the levels of PM2.5 observed in China on the back-trajectory pathways starts to increase from around 1,200 km and worsens as it passes 1,000 km away from the receptor cities. Therefore, in this study we use the average PM2.5 concentration observed in Chinese cities in the pathways identified by the HYSPLIT backward trajectories within 1,000 km from the receptor cities as the LRTT-related variable.

5. We have described in detail the process for finding the optimal spatial panel data model. As mentioned in the main text, we compared the Akaike Information Criteria (AIC) for the spatial error model (SEM) and the spatial autoregressive model (SAR) by testing seven different spatial weight matrices for the SEM and SAR models: queen contiguity, three distance-based (2, 4, and 8 k-nearest-neighbors (KNN)) and three kernel functions (uniform, triangular, and gaussian). We finally selected the SEM models with the 8-KNN spatial weight matrix as the most appropriate models, as these have the lowest AIC.

6. The introduction, conclusion, and policy implications have been rewritten according to the revised analysis results to emphasize the significance and distinction of the current study.

7. We have reorganized the structure of the paper by grouping the analysis into two parts: descriptive and visual analysis, and spatial panel-data analysis.

Reviewer #1:

1. In spatial panel regression, LRTT variables from those Chinese cities are not clear to me. According to the author, LRTT variables are determined by the observed PM2.5 concentration from the source region at the departure time. The author does not discuss the details of the travel time of the PM2.5 in HYSPLIT model. The only information readers can find is the travel time depends on the air mass travel speed and direction, and the maximum of the travel time is five days in HYSPLIT model. That means, at any time t, I do not know for which time the PM2.5 concentration from the source region is observed. 

=>Thank you for your helpful comments. We have provided a detailed explanation on determining PM2.5 levels originating from China. We found that the levels of PM2.5 observed in Chinese cities on the back-trajectory pathways start to increase from around 1,200 km and worsen as they pass 1,000 km away from the receptor cities. Therefore, this study uses the average PM2.5 concentration observed in China in the pathways identified by the HYSPLIT backward trajectories within 1,000 km from the receptor cities as the LRTT-related variable.

2. One of my major concerns is about the endogeneity. It seems that the PM2.5 from Beijing will also affect the PM2.5 concentration in Shenyang by forward trajectories in the Figure 3. If this effect of transboundary sources among source cities is ignored, then the estimation of marginal effects of LRTT from a specific source city on local PM2.5 concentration is not valid. 

=> As the focus of the analysis of this study changes from the source cities in China to the LRTT effects on receptor cities in Korea, we think there would be no endogeneity problems.

3. It would be more helpful if the author can discuss the insignificance of estimated coefficient of LRTT variables for Shenyang. Compared with other three regressions, regional meteorological factors plays an important role in the last regression. The insignificance of coefficient can be caused by other factors. For example, does the PM2.5 from Shenyang have a longer travel time than other cities? 

=> As we changed from forward-trajectory to back-trajectory, the focus of the analysis also changed from LRTT effects from source cities in China to effects on receptor cities in Korea.

4. The author needs to reorganize the structure of the paper. The analysis of identification of source cities by HYSPLIT model should be presented before the econometric model setup. Then the summary statistics of the dependent variables and predictors follows. Figure 2 depicts the patterns of PM2.5 in source cities and South Korea. It can be placed before the test of random effects vs. fixed effects. 

=> We have reorganized the structure of the paper following the reviewer’s helpful suggestions. We grouped the analysis parts into two: descriptive and visual analysis and spatial panel-data analysis. The descriptive and visual analysis includes: 1) analysis of a causal link between high-PM2.5 concentration in Korea and the LRTT-related PM2.5 from China, 2) synoptic meteorological conditions, 3) observed PM2.5 concentrations in China and Korea and back-trajectory pathways, and 4) aerosol optical depth (AOD) and CAMS PM2.5 concentration with wind direction.

5. In Line 206, k is not defined. Be specific about the dimension of explanatory variables. 

=> Following your helpful suggestion, we added specific notations on N, t, and k.

6. In Table 3, some variables’ names are confusing, e.g., P and _. 

=> A revision has been made accordingly.

Reviewer #2:

1. For such spatial dynamic analysis, the modeler shall include all endogenous variables in the model. The author only considered several cities from China and South Korea. It might be the case that cities from Japan and North Korea also contribute to this transboundary dynamic. Simple omission of these possible endogenous variables may lead to severe bias. 

=> We thank you for your insightful comments. This study identified air flows using the back-trajectory model to find the source region of LRTT and found a causal link between the high concentration of PM2.5 in Korea and the LRTT from China. This finding was also confirmed by MODIS-AOD and CAMS satellite image data. This study also analyzed the meteorological conditions favorable for maintenance of the high-PM2.5 episode in Korea, such as air pressure, wind direction and speed, and ambient temperature. We included these meteorological variables into the spatial panel-data model along with the LRTT variable and secondary particles such as sulfur dioxide (SO2) and nitrogen dioxides (NO2).

2. The author has a wrong interpretation of the coefficient rho and lambda. The significance of these coefficients only suggests there exists spatial dynamic among these cities. It does not deliver the information of transboundary direction. 

=> As you argue, the coefficient rho reflects the existence of spatially correlated errors, regardless the transboundary direction. In this study, the LRTT-related PM2.5 level is determined by the average PM2.5 concentration observed in China in the pathways identified by the back-trajectories within 1,000 km from the receptor cities, which includes the information of transboundary direction.

3. The Hausman test to distinguish between fixed effect and random effect is largely applicable to cross sectional setting. I don’t see how the author applied it to spatial panel setting. 

=> We refer to Belotti et al. (2017), which gives instruction on the Hausman test for the spatial panel-data model using Stata, even though it is allowed only for static models.

Reference: 

Belotti et al. (2017) Spatial panel-data models using Stata, The Stata Journal, 17(1), pp. 139–180

4. It is confusing to read the summary statistics in Table 1, where variables appear in both rows and columns. 

=> As we changed the focus of the analysis from LRTT effects from source cities in China to effects on receptor cities in Korea, Table 1 has been changed accordingly.

5. The conjecture of transboundary effects from China to Korea based on Figure 2 is far from serious. 

=> As we changed from forward-trajectory to back-trajectory, we revised Figure 2. In the current version, Figure 3 clearly shows that the regional PM2.5 level in Korea was greatly affected by the LRTT from China.

6. For spatial panel model, the predetermined spatial weight matrix is important. The author shall elaborate this procedure more carefully. 

=> We have described in detail the process for finding the optimal spatial panel data model. As mentioned in the main text, we compared the Akaike Information Criteria (AIC) for the spatial error model (SEM) and the spatial autoregressive model (SAR) by testing seven different spatial weight matrices for the SEM and SAR models: queen contiguity, three distance-based (2, 4, and 8 k-nearest-neighbors (KNN)) and three kernel functions (uniform, triangular, and gaussian). We finally selected the SEM models with the 8-KNN spatial weight matrix as the most appropriate models, as they have the lowest AIC.

7. As a suggestion, the author might want to try a dynamic spatial panel model, see Li, K. (2017). Fixed-effects dynamic spatial panel data models and impulse response analysis. Journal of Econometrics, 198(1), 102-121. 

=> Thank you for your suggestion. Extension of the spatial panel-data model is considered as a future research topic, and the following is mentioned in the conclusion: “extending the statistical model to the spatiotemporal panel data model can be a future research consideration in order to take spatial as well as temporal correlation into account.”

---

## [Decision Letter · Decision Letter 1]

7 Nov 2022

PONE-D-21-36719R1Effects of transboundary PM2.5 transported from China on the regional PM2.5 concentrations in South Korea: A s patial panel-data analysisPLOS ONE

Dear Dr. JUN,

Thank you for submitting your manuscript to PLOS ONE. After careful consideration, we feel that it has merit but does not fully meet PLOS ONE’s publication criteria as it currently stands. Therefore, we invite you to submit a revised version of the manuscript that addresses the points raised during the review process.

We look forward to receiving your revised manuscript.

Kind regards,

Yangyang Xu

Academic Editor

PLOS ONE

Reviewers' comments:

Reviewer's Responses to Questions

**Comments to the Author**

1. If the authors have adequately addressed your comments raised in a previous round of review and you feel that this manuscript is now acceptable for publication, you may indicate that here to bypass the “Comments to the Author” section, enter your conflict of interest statement in the “Confidential to Editor” section, and submit your "Accept" recommendation.

Reviewer #1: (No Response)

Reviewer #2: All comments have been addressed

Reviewer #3: (No Response)

2. Is the manuscript technically sound, and do the data support the conclusions?

Reviewer #1: Yes

Reviewer #2: Yes

Reviewer #3: (No Response)

3. Has the statistical analysis been performed appropriately and rigorously? 

Reviewer #1: Yes

Reviewer #2: N/A

Reviewer #3: No

4. Have the authors made all data underlying the findings in their manuscript fully available?

Reviewer #1: Yes

Reviewer #2: Yes

Reviewer #3: No

5. Is the manuscript presented in an intelligible fashion and written in standard English?

Reviewer #1: Yes

Reviewer #2: Yes

Reviewer #3: Yes

6. Review Comments to the Author

Reviewer #1: I have pointed out several issues in the second round report. Please see the attachment for details.

Reviewer #2: This revision is much improved, compared to the original version. I don't have any further comments or concerns.

Reviewer #3: PONE-D-21-36719R1: statistical review

SUMMARY. This is a study of the effects of Chinese-specific transboundary pollution sources on concentrations of particulate matter in South Korea. The dependent variable is observed in the form of a space-time series of PM concentrations, collected four times per day in the period March 23- April 12 2019, across 396 monitoring stations in Korea. Long-range transboundary transport (LRTT) effects are adjusted by available confounders by a spatial panel regression model. I have several major concerns about this paper, which would require a full revision of the statistical analysis. I also append some specific issues that should be considered.

MAJOR ISSUES:

1. Model specification

1a. Little is said about the monitoring stations in South Korea. I guess they record PM at a selected point in space. These data do not therefore represent the average PM concentrations in an area of study. A discrete spatial model, such as the one considered by the authors, is well suited for spatial areal data. The data of this study are instead geo-statistical data, i.e. they represent pointwise observations of a continuous spatial random field. A geo-statistical model should be considered for the statistical analysis. As an aside comment: I don’t understand how spatial weight matrices were computed. Spatial weight matrices are defined according to a specific neighborhood structure, which in turn requires a spatial tessellation. We don’t have any tessellation here because data are collected at single points in space…

1b. As figure 3 clearly shows, data are temporally correlated, but only spatial auto-correlation was taken into account. Ignoring an important source of autocorrelation may lead to severe bias in the p-value computations

2. Wind direction is a circular variable and requires special care. It seems that the circular support of wind direction has been ignored, as it was included in the model such as the other covariates. This is not correct. When the dependent variable (PM) is linear and a covariate (wind direction) is circular, the regression line is a line wrapped around a cylinder! I would suggest to transform wind direction to a qualitative variable, by clustering value into a finite number of classes. More rigorous approaches should account for the literature of linear-angular regression. As an aside comment: how was average wind direction computed? The equation of a circular mean is different of the standard mean computation of a linear variable.

3. A single spatial autocorrelation parameter for a dataset like this is far from realistic. Spatial dependence tend to vary over time and across space! More generally, model checking is a bit overlooked. What about predictive capability of the model?

4. Although the model relies on the normality assumption of the dependent variable, this assumption was not tested. Is the distribution of PM normal?

5. To be more convincing, the model should include an estimation of pollution from other geographical areas such as North Korea and Japan (not just China...). I guess that data from North Korea are unavailable, and this could be a fatal issue.

SPECIFIC ISSUES

1. Lines 167-168: “Figure 2 indicates that as the westerly wind approaches Korea, it passes through these large cities, increasing the LRTT PM2.5 concentrations”. Figure 2 does not include wind!

2. Figure 2: instead of overlapping two time series, correlation plots would be more informative.

3. Figure 4a is a bit misleading, because of the circular nature of wind direction.

4. The computations of Table 2 depend on a specific spatial neighborhood structure, a priori defined. Which neighborhood structure was considered? See also major issue 1a

7. PLOS authors have the option to publish the peer review history of their article (what does this mean?). If published, this will include your full peer review and any attached files.

Reviewer #1: No

Reviewer #2: No

Reviewer #3: No

---

## [Author Response · Author response to Decision Letter 1]

9 Dec 2022

Reviewer's Responses to Questions

Reviewer #1:

• In line 307, it would be better to clarify the main predictor variables “LRTT of PM2.5 from China”. For some N, at time t, is LRTT of PM2.5 from China defined as the average PM2.5 LRTT of China cities on HYSPLT back-trajectory five days before? I do not find a formal definition of this predictor variable, though the author discusses LRTT of PM2.5 from China on page 9.

 We clearly define LRTT on p. 5 as follows: “this study uses the average PM2.5 concentration observed at the passage time in Chinese cities in the pathways identified by the HYSPLIT backward trajectories within 1,000 km from the receptor cities as the LRTT-related variable in the following analysis, which is obtained every 6 h according to the arrival time at the receptor cities.”

• The author should explain more about Figure 2. One question is that the closest China city is 250km far away from the receptor Korean cities, by Figure 1. How can the author get the weekly average PM2.5 concentrations observed in Chinese cities in the pathways at a distance of 250km?

 We have added a detailed explanation for Figure 2 as follows: “Since we have information on the x-y coordinates of the Chinese cities in the pathways and the receptor cities, and on the PM2.5 concentrations observed in Chinese cities at the passage time, it is possible to calculate the levels of PM2.5 in Chinese cities by distance to the receptor cities in Korea during the study period.”

• In line 243, I guess that the author is discussing Figure 5 instead of Figure 4.

 A revision has been made accordingly.

• In line 167, I cannot see Figure 2 indicates that as the westerly wind approaches Korea, it passes through these large cities.

 We changed the sentence to “Figure 2 indicates that as the air trajectories pass through these large cities, the LRTT PM2.5 concentrations have increased.”

Reviewer #3: 

1a. Little is said about the monitoring stations in South Korea. I guess they record PM at a selected point in space. These data do not therefore represent the average PM concentrations in an area of study. A discrete spatial model, such as the one considered by the authors, is well suited for spatial areal data. The data of this study are instead geo-statistical data, i.e. they represent pointwise observations of a continuous spatial random field. A geo-statistical model should be considered for the statistical analysis.

 We thank you for your insightful comments. Although air quality monitoring stations collect pointwise air pollution levels, they are known to record pollution levels representing an area covering approximately 2 km radius (approximately 12–15 km2). Therefore, we assume that air pollution levels recorded by 396 monitoring stations in Korea represent air quality in urbanized areas.

As an aside comment: I don’t understand how spatial weight matrices were computed. Spatial weight matrices are defined according to a specific neighborhood structure, which in turn requires a spatial tessellation. We don’t have any tessellation here because data are collected at single points in space… 4. The computations of Table 2 depend on a specific spatial neighborhood structure, a priori defined. Which neighborhood structure was considered? See also major issue 1a.

 We constructed a spatial weight matrix (SWM) using the coordinates of the monitoring stations. Spatial weight matrices are divided into contiguity-based and distance-based. Distance-based SWM has three different types: 1) distance band, 2) K-nearest neighbor (KNN), and 3) kernel. In this study, seven different SWMs were constructed and tested: queen contiguity, three different KNN (2, 4, and 8 KNN) and three kernel functions (uniform, triangular, and gaussian). We finally selected 8-KNN as the best SWM by comparing the Akaike Information Criteria (AIC).

1b. As figure 3 clearly shows, data are temporally correlated, but only spatial autocorrelation was taken into account. Ignoring an important source of autocorrelation may lead to severe bias in the p-value computations

 We thank you for your valuable comments. We revised our model by including two lag variables to address serial dependence. We added lag variables of NO2 and SO2, which are key determinants of the regional PM2.5 concentration as the secondary precursor. We added the following sentence on p. 13: “In addition, we added two lag variables of NO2 and SO2, which are key determinants of the regional PM2.5 concentration as the secondary precursor to consider serial dependencies.”

2. Wind direction is a circular variable and requires special care. It seems that the circular support of wind direction has been ignored, as it was included in the model such as the other covariates. This is not correct. When the dependent variable (PM) is linear and a covariate (wind direction) is circular, the regression line is a line wrapped around a cylinder! I would suggest to transform wind direction to a qualitative variable, by clustering value into a finite number of classes. More rigorous approaches should account for the literature of linear-angular regression. As an aside comment: how was average wind direction computed? The equation of a circular mean is different of the standard mean computation of a linear variable.

 We thank you for your valuable suggestion. We replaced the wind direction variable with three qualitative variables representing westerly wind direction by adding the following sentence on p. 13: “For the wind direction, three westerly wind directional dummy variables are included: west-southwest (WSW: 1 if the azimuth wind direction is between 213 and 281 and 0 otherwise), 2) west-northwest (WNW: 282 and 326), and 3) north-northwest (NNW: 327 and 360).”

3. A single spatial autocorrelation parameter for a dataset like this is far from realistic. Spatial dependence tend to vary over time and across space! More generally, model checking is a bit overlooked. What about predictive capability of the model?

 Thank you for your comment. Like other spatial panel data models in the literature, this study also used a spatial panel data model under the assumption that association between the values in nearby locations would be similar over time.

4. Although the model relies on the normality assumption of the dependent variable, this assumption was not tested. Is the distribution of PM normal?

 We thank you for your valuable suggestion. We made a log transformation of the dependent variable (local PM2.5 level) to address the normality issue. Due to the log transformation, the calculation method for the average effects of LRTT from China in Table 4 was also modified.

5. To be more convincing, the model should include an estimation of pollution from other geographical areas such as North Korea and Japan (not just China...). I guess that data from North Korea are unavailable, and this could be a fatal issue.

 As you point out, PM2.5 concentrations in Korea can be affected not only by China, but also by North Korea and Japan. Looking at the wind direction during the study period, the effect from Japan would seem to be negligible. Conversely, North Korea may have an influence, but data is not available. As a result of the spatial panel model analysis, the North Korean wind direction variable (NNW) is neither statistically significant nor has positive values, so the influence from North Korea does not seem to be significant.

SPECIFIC ISSUES

1. Lines 167-168: “Figure 2 indicates that as the westerly wind approaches Korea, it passes through these large cities, increasing the LRTT PM2.5 concentrations”. Figure 2 does not include wind!

 We changed the sentence to “Figure 2 indicates that as the air trajectories pass through these large cities, the LRTT PM2.5 concentrations have increased.”

2. Figure 2: instead of overlapping two time series, correlation plots would be more informative.

 Figure 2 compares PM levels of Korea and LRTT from China, which we think demonstrates visually the relationship between the two variables.

3. Figure 4a is a bit misleading, because of the circular nature of wind direction.

 Figure 4a is a graph showing the average wind direction obtained from the backward trajectory. We believe that this graph helps readers to understand the azimuth of LRTT.

---

## [Decision Letter · Decision Letter 2]

7 Feb 2023

Effects of transboundary PM2.5 transported from China on the regional PM2.5 concentrations in South Korea: A spatial panel-data analysis

PONE-D-21-36719R2

Dear Dr. JUN,

We’re pleased to inform you that your manuscript has been judged scientifically suitable for publication and will be formally accepted for publication once it meets all outstanding technical requirements.

Kind regards,

Yangyang Xu

Academic Editor

PLOS ONE

Additional Editor Comments (optional):

Reviewers' comments:

Reviewer's Responses to Questions

**Comments to the Author**

1. If the authors have adequately addressed your comments raised in a previous round of review and you feel that this manuscript is now acceptable for publication, you may indicate that here to bypass the “Comments to the Author” section, enter your conflict of interest statement in the “Confidential to Editor” section, and submit your "Accept" recommendation.

Reviewer #1: All comments have been addressed

Reviewer #3: All comments have been addressed

2. Is the manuscript technically sound, and do the data support the conclusions?

Reviewer #1: Yes

Reviewer #3: (No Response)

3. Has the statistical analysis been performed appropriately and rigorously? 

Reviewer #1: Yes

Reviewer #3: (No Response)

4. Have the authors made all data underlying the findings in their manuscript fully available?

Reviewer #1: Yes

Reviewer #3: (No Response)

5. Is the manuscript presented in an intelligible fashion and written in standard English?

Reviewer #1: Yes

Reviewer #3: (No Response)

6. Review Comments to the Author

Reviewer #1: Please take a look at the attachment.

Reviewer #3: (No Response)

7. PLOS authors have the option to publish the peer review history of their article (what does this mean?). If published, this will include your full peer review and any attached files.

Reviewer #1: No

Reviewer #3: No

---

## [Editor Report · Acceptance letter]

23 Feb 2023

PONE-D-21-36719R2 

Effects of transboundary PM_2.5_ transported from China on the regional PM_2.5_ concentrations in South Korea: A spatial panel-data analysis 

Dear Dr. Jun:

I'm pleased to inform you that your manuscript has been deemed suitable for publication in PLOS ONE. Congratulations! Your manuscript is now with our production department. 

Kind regards, 

on behalf of

Dr. Yangyang Xu 

Academic Editor

PLOS ONE